# Fungal Diversity of Deteriorated Sparkling Wine and Cork Stoppers in Catalonia, Spain

**DOI:** 10.3390/microorganisms8010012

**Published:** 2019-12-19

**Authors:** Ernesto Rodríguez-Andrade, Alberto M. Stchigel, Josep Guarro, José F. Cano-Lira

**Affiliations:** Mycology Unit, Medical School and IISPV, Universitat Rovira i Virgili (URV), Sant Llorenç 21, Reus, 43201 Tarragona, Spain; dc.ernesto.roan@outlook.com (E.R.-A.); josep.guarro@urv.cat (J.G.); jose.cano@urv.cat (J.F.C.-L.)

**Keywords:** ascomycota, *cava*, cork taint, fungi, sparkling, spoilage, stoppers, wine

## Abstract

Filamentous fungi are rarely reported as responsible for spoiling wine. Cork taint was detected in sparkling wine; therefore, we investigated fungal contamination as a possible cause of organoleptic alteration. Spoiled wine was filtered and membranes were plated onto potato dextrose agar (PDA). The cork stoppers used for sealing bottles were cut and also plated onto PDA. Fungal strains were phenotypically characterized and molecularly identified by sequencing of a fragment of the 28S nrRNA gene (LSU) and (occasionally) by other additional molecular markers. Twenty-seven strains were isolated and sixteen species were identified, all of them belonging to the phylum Ascomycota. The fungi isolated from wine were three species of *Aspergillus* section *Nidulantes*, a species of *Penicillium* section *Exicaulis* and *Beauveria bassiana*. *Candida patagonica* was isolated from both sort of samples, and the fungi isolated from cork stoppers were *Altenaria alternata* and *Cladosporium cladosporioides*. Surprisingly, most of the taxa recovered from the cork stoppers and/or wine were new to the science: a new genus (*Dactylodendron*) and seven new species belonging to the genera *Cladophialophora*, *Dactylodendron*, *Kirschsteiniothelia*, *Rasamsonia*, and *Talaromyces*. Future studies could let us know if these fungi would be able to produce compounds responsible for cork taint.

## 1. Introduction

Sparkling wine is one of the most economically important wine varieties in southern Europe. It is produced by the “champenoise” method, which consists of two steps: a primary alcoholic fermentation, in which the grape must is transformed to the wine base (cuvée); and a second alcoholic fermentation after the addition of sucrose, selected yeasts, and bentonite to the base wine, which is then bottled, closed with a metal cap or a cork stopper, and allowed to age in cellars for a longer period of time (at least 12 months for French champagne and 9 months in the case of the Spanish—mostly Catalonian— “cava”) [1]. During fermentation, a certain diversity of environmental microorganisms, mainly bacteria and fungi, can produce organoleptic alterations that render the wine undrinkable. Some of these fungi can be present on the cork stoppers and/or be acquired by exposure of the must to bio-aerosols, perhaps because of poor environmental microbiological control at the cellar. Cork taint is a musty or mouldy off-odor in wine often caused by the presence of 2,4,6-trichloroanisole (2,4,6-TCA) among other chemical compounds [2], and between 0.5 and 7% of wines can be affected by cork taint. It is estimated that cost of cork-related wine spoilage can exceed several billions of dollars per year [3,4]. The metabolic effect of fungi living on cork in the production of 2,4,6-TCA has been described [5]. Among several fungi recovered from agglomerate cork stoppers, *Acremonium strictum*, *Chrysonilia sitophila*, *Cladosporium oxysporum*, *Fusarium oxysporum*, *Paecilomyces viridis*, *Penicillium chrysogenum*, *Trichoderma longibrachiatum*, *Trichoderma viride*, and *Verticillium psalliotae* have displayed such an effect [5].

Surprisingly, little is known about the nature of the wine spoilage fungi, but there have been some reports about these organisms being isolated during and at the end of fermentation, such as *Cladosporium cucumerinum*, *Cryptococcus tephrensis*, *Hanseniaspora thailandica*, *Schizosaccharomyces japonicas*, and *Sporobolomyces coprosmae* [6,7,8,9,10,11,12,13,14]. Most of those studies focused on microorganisms present on cork stoppers. Some filamentous fungi have been reported on this substrate, such as *Alternaria alternata*, *Armillaria mellea*, *Aspergillus* spp., *Aureobasidium pullulans*, *Cladosporium* spp., *Fusarium* spp., *Mucor* spp., *Neurospora sithophila*, *Penicillium* spp., *Rhizopus arrhizus*, *Scopulariopsis candida*, and *Trichoderma* spp., and also yeasts such as *Debaryomyces hansenii*, *Lachancea thermotolerans*, *Rhodotorula* spp., *Sporidiobolus johnsonii*, several species of *Saccharomyces*, *Tausonia pullulans*, and *Trichomonascus ciferri* [15,16,17,18,19,20,21,22,23,24,25,26,27]. Interestingly, other yeasts, such as *Bullera* sp., *Cutaneotrichosporon mucoides*, *Cryptococcus albidus*, *D. hansenii*, *Rhodosporidium kratochviloae*, *Rhodotorula* spp., *Sporidiobolus* spp., and *Wickerhamomyces anomalus* have all been isolated during the manufacturing process of the cork stoppers [28]. In addition, the sordariaceous mold *Zopfiella ebriosa* has been found on cork stoppers exposed to wine [29].

We recently had the opportunity to study the fungal biota associated to wine deterioration when a local winery located in Sant Sadurní d’Anoia (Barcelona province, Spain) detected cork taint in some bottles of sparkling wine during an inspection of the cellars at its historic vineyards. We conducted a study to detect, isolate, and identify the fungi involved in the production of this sort of flavor alteration.

## 2. Materials and Methods

### 2.1. Fungal Isolation

Samples of sparkling wine and cork stoppers were obtained from a cellar in Sant Sadurní d’Anoia, Barcelona province, Spain. Approximately, 500 bottles of sparkling wine from five different batches were opened in situ to obtain a representative number of negative controls (without organoleptic alteration) and a panel of four experts detected taste defects in any samples that had a musty or mouldy off-odor and/or flavor. A total of 54 bottles of sparkling wine sealed by cork stoppers (15 negative control and 39 with deteriorated wines) were selected and processed. A sample of 100 mL of sparkling wine was filtered through a filter membrane of 0.45 µm diameter (Millipore SA, Molsheim, France). After filtering, the membrane was plated onto a 90 mm diameter Petri dish containing potato dextrose agar (PDA; Pronadisa, Madrid, Spain) plus 50 mg/L L-chloramphenicol. The Petri dishes were incubated at 25 °C for a time period ranging from 4 weeks to 2 months in darkness, and examined under a stereomicroscope to observe any production of mold colonies with reproductive structures. If bacteria and/or yeasts develop on the culture medium, these could be recognized by mucous to buttery colonies of reduced diameter, and by the absence of hyphae; also, slide mountings on water seen under bright field microscope allows detection of the bacterial/yeast cells. The cork stoppers were cut into small pieces using a sterile disposable scalpel and plated onto 90 mm diameter Petri dishes containing PDA, which were incubated in the same way as described above. For both sorts of samples, fungal structures from selected colonies (representative of all the morphological varieties) were transferred to 50-mm diameter Petri dishes containing PDA using a sterile insulin-type needle and incubated in the same conditions to obtain pure cultures.

### 2.2. Phenotypic Characterization of the Fungal Strains

For the isolates of *Rasamsonia* and *Talaromyces*, suspensions of conidia were prepared in a semi-solid agar (0.2% agar, 0.05% Tween 80) [30] and inoculated in three equidistant points onto 2% malt extract agar (MEA; Difco Inc., Detroit, USA), oatmeal agar (OA) [30], Czapek yeast extract agar (CYA) [31], yeast extract sucrose agar (YES) [32], creatine sucrose agar (CREA) [32], dichloran 18% glycerol agar (DG18) [33], and cork (cut in slices by a scalpel, placed into appropriate containers, and sterilized three times in alternative days at 121 °C during 15 minutes) onto tap water agar (TWA; 1.5% agar in tap water) into disposable Petri dishes of 90 mm diameter, and incubated at 25 °C in darkness after 14 days. Cultures on CYA were also incubated at different temperatures (5, 15, 25, 30, 35, and 37 °C) to determine the minimum, optimal, and maximum temperatures of growth. The rest of the isolates were cultured and studied onto MEA, OA, PDA, and TWA with pieces of sterile cork, incubated at 25 °C in darkness after 14 days. The determination of the cardinal temperatures of growth of these strains was determined on PDA. Color notations in parentheses are from Kornerup and Wanscher [34]. The characterization and measurements of fungal structures were performed in water and 60% lactic acid from slide cultures by using the culture media cited before. Photographs were taken by a Zeiss Axio Imager M1 light microscope (Zeiss, Oberkochen, Germany) with a DeltaPix Infinity X digital camera, using Nomarski differential interference contrast. The samples for scanning electron microscopy (SEM) were processed according to Figueras and Guarro [35], and SEM micrographs were taken at 15 keV with a Jeol JSM 840 microscope. The taxonomic descriptions and names of the fungal novelties were introduced into MycoBank (www.mycobank.org) [36].

### 2.3. DNA Extraction, Amplification, and Sequencing

Total DNA, extracted by the modified protocol of Marimon et al. [37], was used to amplify and sequence a fragment of the 28S nrRNA gene (LSU) using the primer pair LR0R [38] and LR5 [39] for all isolates. For the phylogeny of the isolates of *Cladophialophora*, *Rasamsonia,* and *Talaromyces*, the following molecular markers were amplified and sequenced: i.e., ribosomal internal transcribed spacers (ITS) (ITS5/ITS4) [40] and fragments of the beta-tubulin (*BenA*) (Bt2a/Bt2b) [41]; calmodulin (*CaM*) (Cmd5/Cmd6) [42] and RNA polymerase II subunit 2 (*rpb*2) (RPB2-5F/RPB2-7cR) [43] genes. Sequencing of the amplicons was made in both directions with the same primer pair used for amplification at Macrogen Europe (Macrogen Inc., Amsterdam, The Netherlands). The consensus sequences were obtained using the SeqMan software v. 7 (DNAStar Lasergene, Madison, WI, USA). Sequences generated in the present work were deposited in GenBank (Table 1).

### 2.4. Preliminary Identification and Phylogenetic Analysis

Preliminary molecular identification was carried out by comparing of the LSU sequences of our isolates with those of the type or reliable GenBank reference strains using the Basic Local Alignment Search Tool (BLAST; https://blast.ncbi.nlm.nih.gov/Blast.cgi). A maximum level of identity (MLI) of ≥98% was considered to allow for species-level identification. MLI values < 98% provided identification only at genus level. For identification of species of *Aspergillus* and *Penicillium*, sequences of a fragment of *BenA* gene were used. To determine the phylogenetic placement of all our isolates, an LSU tree was built. Additionally, three trees with a combined data set were built to distinguish among the species of *Talaromyces* section *Trachyspermi* (by using the combined dataset ITS-*BenA*-*CaM*-*rpb*2), the species of *Rasamsonia* (ITS-*BenA*-*CaM*), and the species of *Cladophialophora* (ITS-LSU-*BenA*). *Candida bituminiphila* and *Candida patagonica* for LSU tree, *Talaromyces rademirici* and *Talaromyces dendriticus* for *Talaromyces* section *Trachyspermi* tree, *Talaromyces flavus* and *Trichocoma paradoxa* for *Rasamsonia* spp. tree, and *Exophiala oligosperma* and *Exophiala exophialae* for *Cladophialophora* spp. tree were used as out-groups. For sequence alignment and to perform the maximum-likelihood (ML) and Bayesian-inference (BI) phylogenetic analyses, we followed the methodology described by Valenzuela-Lopez et al. [44]. The final matrices used for phylogenetic analysis were deposited in TreeBASE (www.treebase.org; accession number: S23148).

### 2.5. Growing at Different Ethanol Concentrations

Strains from cork stoppers and sparkling wine were grown on test tubes with 5 mL of 2% malt extract in tap water supplemented with different amounts of ethyl alcohol to reach 5, 10, 15, and 20% v/v final concentration. The tubes were closed with plastic caps, hermetically sealed by parafilm^®^, and incubated at 15 °C for up to 13 months in darkness without agitation, trying to simulate the method employed for resting/aging of wine. The tubes were examined every month for fungal growth. If growth was absent, 0.1 mL of the broth was plated onto PDA, and incubated during 2 weeks in darkness at 25 °C to confirm absence of fungal growth.

## 3. Results

### 3.1. Fungal Diversity of Cork Stoppers and Sparkling Wine Samples

None of the negative controls of sparkling wine developed fungal colonies. On the other hand, 24 out of 39 odor/flavor altered samples developed bacterial, yeasts, and/or mold colonies. All of the cork stopper samples developed fungal colonies. A total of 27 ascomycetes, representing all the morphological variability of the fungal colonies produced, were isolated from cork stoppers and from sparkling wine. Five of them were identified as *Talaromyces* spp., four as *Kirschsteiniothelia* spp., two as *Rasamsonia* spp. and one as *Cladophialophora* sp., three other strains belonged to an unknown arthrosporate fungus. Several *Aspergillus* spp. were recovered from sparkling wine: i.e., *Aspergillus aureolatus*, *Aspergillus jensenii,* and *Aspergillus puulaauensis* (all of them belonging to the section *Nidulantes*). *Penicillium corylophilum* (section *Exilicaulis*) and several other fungi were identified from both sparkling wine and cork stoppers. *Alternaria alternata* and *Cladosporium cladosporioides* were identified on cork stoppers, but *Beauveria bassiana* and *Candida patagonica* were only found in sparkling wine.

### 3.2. Molecular Phylogeny

The first phylogenetic study included 64 LSU sequences, with a total of 517 characters including gaps, from which 273 were parsimony informative. The ML analysis was congruent with the BI analysis, both displaying a similar topology. In the LSU tree, our fungal isolates were distributed across two main clades (Figure 1), the first (100% BS / 1 PP), corresponding to the filamentous Ascomycota, included 24 of our isolates, and the second (100% BS / 1 PP), corresponding to the class Saccharomycetales (true yeasts), included the other isolates (three). The first main clade divided into six subclades: A (unsupported, including 13 isolates), corresponding to the order Eurotiales; B (100% BS / 1 PP, three isolates), representing the family Eremascaceae (of the order Onygenales); C (100% BS / 1 PP, one isolate), grouped the family Herpotrichiellaceae (order Chaetothyriales); D (unsupported, five isolates), which included the family Kirschteiniotheliaceae (sister clade D1; 100% BS / 1 PP) and the family Pleosporaceae (sister clade D2; 100% BS / 1 PP) (both pertaining to the order Pleosporales); E (100% BS / 1 PP, one isolate), with the family Cladosporiaceae (order Capnodiales); and F (100% BS / 1 PP, one isolate), with the family Cordycipitaceae (order Hypocreales). Subclade A has three well-supported sister clades, representing the genera *Rasamsonia* and *Talaromyces* (sister clade A1; 90% BS / 0.99 PP), *Penicillium* (sister clade A2; 100% BS / 1 PP), and *Aspergillus* (sister clade A3; 88% BS / 0.99 PP). In this context, seven of our isolate formed three well-supported branches within the sister clade A1: two within the genus *Talaromyces* and the third near to the species of *Rasamsonia*, but not closely related to any of the known species. Within the sister clade A2, two isolates grouped with *Penicillium corylophilum*. The sister clade A3 has two isolates placed together with *Aspergillus aureolatus* and another two together with *Aspergillus puulaauensis* and *Aspergillus jensenii*. Subclade B includes three isolates related to *Arthrographis pinicola* and *Eremascus albus* (Onygenales). Subclade C groups different species of *Cladophialophora*, the isolate FMR 16667 being phylogenetically closely related to the type strain of *Cladophialophora mycetomatis*. Sister clade D1 (*Kirschsteiniothelia* spp. and *Dendryphiopsis* spp.) includes four of our isolates, grouped within two fully supported branches. Sister clade D2 includes FMR 15666 and *Alternaria alternata*. Subclade E includes FMR 15660, *Cladosporium silenes*, *C. cladosporioides*, and *C. grevilleae*. Subclade F includes FMR 16669, *Beauveria brongniartii,* and *B. bassiana*. Finally, in the second main clade (Saccharomycetales), three of our isolates and *Candida patagonica* were placed into a well-supported sister branch (100% BS / 1 PP).

Three additional phylogenies allowed the taxonomy of *Talaromyces*, *Rasamsonia*, and *Cladophialophora* to be resolved. The first (ITS, *BenA*, *CaM*, and *rpb*2) clarified the relationships among the species of *Talaromyces* section *Trachyspermi*, which included five of our isolates (Figure 2). The final concatenated dataset was obtained using both ML and Bayesian analyses. It contained 29 taxa with a total of 2270 characters including gaps (515 of them for ITS, 376 for *BenA*, 527 for *CaM*, and 852 for *rpb*2), 728 of which were parsimony informative (128 of them for ITS, 145 for *BenA*, 212 for *CaM*, and 243 for *rpb*2). The datasets did not conflict with the tree topologies for the 70% reciprocal bootstrap trees, which allowed the four genes to be combined for the multi-locus analysis. The support values were only slightly different between these two analyses. Within the main clade, corresponding to *Talaromyces* section *Trachyspermi* (100% BS / 1 PP), three of our isolates were placed in a distinct branch (100% BS / 1 PP), related with *T. affinitatimellis* and *T. basipetosporus*; the other two isolates were located within another branch (100% BS / 1 PP) of a well-supported terminal clade (100% BS / 0.99 PP), which also included *Talaromyces brasiliensis*.

The second additional phylogenetic analysis was performed (ITS, *BenA*, and *CaM*) to resolve the taxonomical placement of two of our isolates between the genera *Talaromyces* and *Rasamsonia* (Figure 3). The final concatenated dataset contained 15 sequences with a total of 1657 characters including gaps (707 of them for ITS, 392 for *BenA*, and 558 for *CaM*), 424 of which were parsimony informative (120 of them for ITS, 115 for *BenA*, and 189 for *CaM*). The ML analysis showed a similar topology and was congruent with the Bayesian analysis. The phylogenetic tree distinguished a main clade corresponding to the genus *Rasamsonia* (92% BS / 1 PP), which was divided in three subclades; our two isolates were located within one of them (83% BS / – PP) in a well-supported branch related to *R. cylindrospora*, *R. brevistipitata*, *R. columbiensis*, and *R. pulvericola*.

ITS-LSU-*BenA* analysis included sequences from 23 taxa of *Cladophialophora*, with a total of 1554 characters including gaps (615 of them for ITS, 562 for LSU, and 377 for *BenA)*, 377 of which were parsimony informative (209 of them for ITS, 55 for LSU, and 113 for *BenA*). The topologies of both ML and Bayesian analyses showed similar topologies and so were congruent. In the phylogenetic tree (Figure 4), a main clade corresponding to *Cladophialophora* spp. (100% BS / 1 PP) was obtained. Within this clade, a terminal branch (100% BS / 1 PP) included *Cladophialophora mycetomatis* and FMR 16667.

### 3.3. Alcohol Tolerance

All the isolates tested displayed good to excellent growth at 5% v/v ethanol, but failed to grow at higher concentrations of alcohol.

### 3.4. Taxonomy

#### 3.4.1. Subclade A: Eurotiales

Because our strains FMR 16662, FMR 16663, and FMR 16667 form a separate branch into a terminal clade including *T. basipetosporus* and *T. affinitatimellis*, and strains FMR 15656 and FMR 15664 form another independent branch within a terminal clade that also includes *T. brasiliensis* (Figure 2), and because all of them display enough phenotypic and phylogenetic differences with respect to the other species of *Talaromyces* section *Trachispermi* to be considered two new species, we propose the erection of *Talaromyces speluncarum* and *Talaromyces subericola* as follows:

***Talaromyces speluncarum* Rodr.-Andr., Cano et Stchigel, sp. nov**. MycoBank MB 830606. (Figure 5).

Etymology: From Latin speluncarum, of caves, the place where the wine is aged.

Diagnosis: *Talaromyces speluncarum* falls into a terminal clade which also includes *T. basipetosporus* and *T. affinitatimellis* (Rodríguez-Andrade et al., in press). *Talaromyces speluncarum* is easily distinguishable by its spinose enteroblastic conidia arising from phialides (smooth-walled and of retrogressive conidiogenesis in *T. basipetosporus* and *T. affinitatimellis*).

Type: Spain: Barcelona province: Sant Sadurní d’Anoia, from sparkling wine, 3 Apr. 2011, J. F. Cano-Lira & A. M. Stchigel (CBS H-23372 – holotype; CBS 143844 = FMR 16671 – ex-type cultures; LSU sequence GenBank LS453296).

Description: Colonies on CYA: reaching 2–3 mm diameter after 2 weeks at 25 °C, slightly elevate, velvety to floccose, margins regular, yellowish white (4A2), exudate absent, sporulation sparse; reverse greyish orange (5B4), diffusible pigment absent. Mycelium: abundant, composed of subhyaline to pale brown, smooth- and thin-walled, septate hyphae, 1.5–2 µm wide. Conidiophores: mostly biverticillate, less frequently monoverticillate or irregularly verticillate, stipitate, smooth- and thin-walled, 17–32 × 2.5–3 µm, bearing 2–3 branches at the top; branches 1-septate or non-septate, hyaline, smooth- and thin-walled, cylindrical, 8–17 × 2.5–3 µm, bearing 1–4 conidiogenous cells at the top. Conidiogenous cells: phialidic, smooth- and thin-walled, mostly cylindrical and slightly slender toward the apex, 7–10 × 1.5–2.5 µm, frequently with a hyaline, broad, and flattened collarette. Conidia: enteroblastic, one-celled, pale greenish when young, mid brown when mature, spinose to verrucose, globose, 3–4 µm diameter, in basipetal chains of up to 30. Sexual morphology: not observed.

Colonies on MEA: reaching 12–13 mm diameter after 2 weeks at 25 °C, slightly elevate, velvety to floccose, irregular margins, greyish orange (5B6) at center, pale yellow (4A4) to the edge, exudate abundant, dark orange (5A8), sporulation abundant; reverse dark orange (5A8), diffusible pigment golden yellow (5B7). Colonies on DG18: reaching 7–8 mm diameter after 2 weeks at 25 °C, slightly elevated, velvety to floccose, yellowish-white (4A2) at center, white (4A1) to the edge, exudates absent, sporulation sparse; reverse pale orange (5A3), diffusible pigment absent. Colonies on OA: reaching 9–10 mm diameter after 2 weeks at 25 °C, elevated, velvety to floccose, margins entire, greyish yellow (4C4), exudates absent, sporulation abundant; reverse, diffusible pigment absent. Colonies on YES: reaching 5–6 mm diameter after 2 weeks at 25 °C, elevated, velvety, irregular margins, white (4A1), exudate absent, sporulation absent; reverse greyish orange (5B4), diffusible pigment absent. Colonies on TWA with sterile cork: olive brown (4D4), exudate absent, sporulation abundant; reverse olive brown (4D4), diffusible pigment absent. The fungus does not grow on CYA at 30 °C or on CREA at 25 °C.

Other specimens examined: Spain: Barcelona province: Sant Sadurní d’Anoia, from sparkling wine, 3 Apr. 2011, J. F. Cano-Lira & A. M. Stchigel (FMR 16662 and FMR 16663).

***Talaromyces subericola* Rodr.-Andr., Cano et Stchigel, sp. nov**. MycoBank MB 830607. (Figure 6)

Etymology: From Latin suber, cork, because of the origin of the fungus.

Diagnosis: *Talaromyces subericola* differs from *T. brasiliensis* [45] in faster growing rates of the colonies on all culture media tested, and by the production of smooth-walled to verruculose conidia (coarsely verrucose in *T. brasiliensis*).

Type: Spain: Barcelona province: Sant Sadurní d’Anoia, from sparkling wine, 3 Apr. 2011, J. F. Cano-Lira & A. M. Stchigel (CBS H-23366 – holotype; CBS 144322 = FMR 15656 – ex-type cultures; LSU sequence GenBank LS453299).

Description: Colonies on CYA: reaching 30–32 mm diameter after 2 weeks at 25 °C, slightly elevated, floccose, sulcate, margins entire, pale yellow (4A3), and dawn grey (4D1) at center, white (4A1) to the edge, exudate absent, sporulation sparse; reverse violet brown (10F7) at center and pale brown (7D7) to the edge, diffusible pigment absent. Mycelium: abundant, composed of subhyaline, smooth- and thin-walled, septate, anastomosing hyphae, of 2-µm wide. Conidiophores: biverticillate, short-stipitate, smooth- and thin-walled, 30–45 µm × 2–3 µm; branches hyaline, mostly non-septate, smooth- and thin-walled, 2–3 by stipe, bearing 2–4 conidiogenous cells at the top, cylindrical, 12–20 µm × 2–3 µm. Conidiogenous cells: phialidic, smooth- and thin-walled, mostly cylindrical and occasionally slightly slender toward the apex, 7–10 × 2–3 µm. Conidia: enteroblastic, one-celled, broadly ellipsoidal to globose, pale green to pale brown when young but soon becoming mid brown, smooth-walled but verruculose with the age, 3 µm diameter, in basipetal chains of up to 20. Sexual morphology: not observed on the culture media tested.

Colonies on MEA: reaching 27–30 mm diameter after 2 weeks at 25 °C, flat, floccose, sulcate, irregular margins, yellowish white (4A2), exudate absent, sporulation sparse; reverse greyish yellow (4B5) at center and pale yellow (4A3) to the edge, diffusible pigment absent. Colonies on DG18: reaching 13–14 mm diameter after 2 weeks at 25 °C, flat, velvety to floccose, yellowish white (4A2), exudates absent, sporulation sparse; reverse golden yellow (5B8), diffusible pigment absent. Colonies on OA: reaching 44–46 mm diameter after 2 weeks at 25 °C, flat, velvety, margins entire, olive brown (4F4), exudates absent, abundant sporulation; reverse, diffusible pigment absent. Colonies on YES: reaching 31–32 mm diameter after 2 weeks at 25 °C, slightly elevated, velvety, irregular margins, yellowish white (4A2) at center, and white (5A1) to the edge, exudate absent, sporulation sparse; reverse orange (5A7), diffusible pigment absent. Colonies on CREA: reaching 24–26 mm diameter after 2 weeks at 25 °C, moderately elevated, cottony, grey (5B1), exudate absent, sporulation absent, and acid production absent. Colonies on TWA with sterile cork: olive brown (4E4), exudate absent, sporulation abundant; reverse olive brown (4E4), diffusible pigment absent. Colonies on CYA: at 30 °C, reaching 16–18 mm. The fungus does not grow on CYA at 37 °C.

Other specimens examined: Spain: Barcelona province: Sant Sadurní d’Anoia, from sparkling wine, 3 Apr. 2011, J. F. Cano-Lira & A. M. Stchigel (FMR 15664).

Because FMR 16670 and FMR 16675 were placed together in a branch (Figure 3) that is phylogenetically distant from other species of the genus *Rasamsonia*, and due to their phenotypic differences with the other species, *Rasamsonia frigidotolerans* is therefore proposed as a new species.

***Rasamsonia frigidotolerans* Rodr.-Andr., Cano et Stchigel, sp. nov**. MycoBank MB 830608. (Figure 7)

Etymology: From Latin frigus-, cold, and -tolerans, tolerant, in reference to its ability to grow at relatively low temperatures.

Diagnosis: Differing notably from other species of the genus [46,47,48,49,50] by the absence of growth on CYA at 30 °C (after one week incubation, greater than 5 mm diameter in the other species), and by the production of globose conidia (ellipsoidal, ovoid to cylindrical in the rest of the species), with the exception of *R. pulvericola*. However, *R. frigidotolerans* can be easily differentiated from *R. pulvericola* by its production of smooth-walled stipes and branches (verrucose in *R. pulvericola*), and because the conidia are connected by disjunctors (absent in the rest of the species of the genus).

Type: Spain: Barcelona province: Sant Sadurní d’Anoia, from sparkling wine, 3 Apr. 2011, J. F. Cano-Lira & A. M. Stchigel (CBS H-23373 − holotype; CBS 143845 = FMR 16675 − ex-type cultures; LSU sequence GenBank LS453294).

Description: Colonies on CYA: reaching 3–4 mm diameter after 2 weeks at 25 °C, elevated, velvety to floccose, sulcate, margins irregular, pale yellow (4A3), exudate absent, sporulation sparse; reverse pale yellow (4A5), diffusible pigment absent. Mycelium: abundant, composed of hyaline, smooth- and thin-walled when young, becoming slightly verrucose with the age, and septate hyphae, 2 µm wide. Conidiophores: mostly monoverticillate, sometimes biverticillate, stipitate, smooth- and thin-walled, 12–50 µm × 1.5–2.5 µm, bearing up to 2 branches at the top; branches hyaline, non-septate, smooth- and thin-walled, 10–15 µm × 1.5–2 µm. Conidiogenous cells: phialidic, smooth- and thin-walled, in pressed verticils of 2–5 at the top of the stipe or of the branches, slender toward the apex, cylindrical, 8–14 µm × 1.5–2 µm. Conidia: enteroblastic, one-celled, smooth-walled, pale brown, globose, 1–2 µm diameter, in basipetal chains of up to 20 and connected by disjunctors. Sexual morphology: not observed on the culture media tested.

Colonies on MEA: reaching 22–23 mm diameter after 2 weeks at 25 °C, slightly elevated, velvety, irregular margins, greyish yellow (4B4), exudate absent, sporulation abundant; reverse olive brown (4D8), diffusible pigment absent. Colonies on DG18: reaching 5–7 mm diameter after 2 weeks at 25 °C, elevated, velvety to floccose, pale yellow (4A3), exudates absent, sporulation sparse; reverse yellowish orange (4A6), diffusible pigment absent. Colonies on OA: reaching 10–11 mm diameter after 2 weeks at 25 °C, flat, velvety, margins entire, greyish yellow (4C7), exudates absent, abundant sporulation; reverse, diffusible pigment absent. Colonies on YES: reaching 7–9 mm diameter after 2 weeks at 25 °C, elevated, velvety to floccose, irregular margins, yellowish white (4A2) at center, and greyish yellow (4B3) at the edge, exudate absent, sporulation abundant; reverse orange yellow (4B8), diffusible pigment absent. Colonies on TWA with sterile cork: greyish yellow (3B7), exudate absent, sporulation abundant; reverse greyish yellow (3B7), diffusible pigment absent. Growth on CYA at 30 °C and on CREA at 25 °C: absent. Minimum, optimal, and maximum temperature of growth: 15 °C, 25 °C, and 35 °C, respectively.

Other specimens examined: Spain: Barcelona province: Sant Sadurní d’Anoia, from sparkling wine, 3 Apr. 2011, J. F. Cano-Lira & A. M. Stchigel (FMR 16670).

#### 3.4.2. Subclade B: Onygenales

Because *Arthrographis pinicola* is phylogenetically placed far from the type species of the genus *Arthrographis* (*Arthrographis kalrae*), which is located within the family Eremomycetaceae (class Dothideomycetes), and because of the morphological differences with *Eremascus albus*, we introduce the new genus *Dactylodendron* into the family Eremascaceae (order Onygenales, class Eurotiomycetes) (Figure 1), and design *Dactylodendron pinicola* (formerly *Arthrographis pinicola*) as the type species of the genus.

***Dactylodendron Stchigel*, Rodr.-Andr. et Cano, gen. nov**. MycoBank MB 827858.

Etymology: From Greek δάχτυλο-, finger, and -δένδρον, tree, due to the aspect of the conidiophores.

Diagnosis: Recognized by its hyaline, hyphae-like, successively branched conidiophores, or short-stalked conidiophores ending in a verticilate arrangement of fertile branches. In both cases, fertile branches produce hyaline, cylindrical, or cuboid arthroconidia.

Type species: *Dactylodendron pinicola* (Sigler & Yamaoka) Rodr.-Andr., Cano et Stchigel. Mycobank MB 827859.

Description: Colonies: slow-growing at room temperature, always with shades of yellow. Conidiophores semi-macronematous, hyphae-like, single or grouped in discrete dome-shaped or floccose conidiomata, erect, successively branched, or short-stalked, ending in a verticilate arrangement of fertile branches, fertile branches eventually producing arthroconidia. Arthroconidia: hyaline, smooth-walled, usually truncate at both ends, cylindrical or cuboid, produced by transverse septation in basipetal order, separated very late by schizolytic secession from the conidiogenous branches, without disjunctors or separating cells. Chlamydospores: occasionally seen. Sexual morphology: not observed.

***Dactylodendron pinicola* (Sigler & Yamaoka) Rodr.-Andr., Cano et Stchigel, comb. nov**. MycoBank MB 827859.

Basionym: Arthrographis pinicola Sigler & Yamaoka, Canadian Journal of Microbiology 36: 78 (1990) [MycoBank MB 126499].

Description: Hyphae: septate and hyaline, (0.5–) 0.8–2.5 µm wide, bearing narrow conidiophores which branch repeatedly to form floccose conidiomata. The fertile branches are initially sparsely septate and of uniformly narrow diameter, but, as arthroconidial development begins, the apical region broadens and septation occurs in basipetal sequence to form many small cells. Arthroconidia secede by schizolysis, often remaining connected in chains of 3 to 4, which then undergo further schizolysis. There are no disjunctors or separating cells. Mature arthroconidia are smooth, hyaline, tan in mass, cylindrical, but often broader than long, 1.5–4.0 × 1.5–2.5 µm wide. Teleomorph was not observed. No yeast stage was observed.

Notes: The habitat reported for such fungus is wood of *Pinus contorta* var. *latifolia*, especially in galleries and adult beetles of Ips latidens, and from galleries of *Dendroctonus ponderosae* in Alberta, Canada.

Because the genus *Dactylodendron* was placed into the subclade B (Figure 1), and was divided into three different sister branches comprising *D. pinicola*, FMR 16678, FMR 15658, and FMR 16677, and because of the morphological differences among them, the new species *Dactylodendron ebriosum* and *Dactylodendron pluriseptatum* are proposed as follows:

***Dactylodendron ebriosum* Rodr.-Andr., Cano et Stchigel, sp. nov**. MycoBank MB827862. (Figure 8)

Etymology: From Latin ebrios, drunk, due to the habitat of this fungus.

Diagnosis: Morphologically resembling to *Staheliella nodosa*, because of the aspect of the upper part of the conidiophores and the sort of conidiogenesis. However, *S. nodosa* clearly differs from *D. ebriosum* because the former is a dematiaceous fungus with longer and wider conidiophores, which also display a percurrent development, bearing several fertile loci along them (the unique fertile part of *D. ebriosum* conidiophores are at the tip of the same).

Type: Spain: Barcelona province: Sant Sadurní d’Anoia, from a cork stopper, 03 June 2011, J. F. Cano-Lira & A. M. Stchigel (CBS H-23367 − holotype; CBS 144321 = FMR 15658 − ex-type cultures; LSU sequence GenBank LT985880).

Description: Colonies on PDA: reaching 10–11 mm diameter after 2 weeks at 25 °C, velvety to floccose, slightly elevated, margins slightly irregular, reddish yellow (4A6) at center, vivid yellow (3A8) and white (3A1) to the edge, exudate absent, sporulation sparse; reverse yellowish-white (3A2), diffusible pigment absent. Mycelium: composed of septate, hyaline, smooth- and thin-walled, 1–2 µm wide hyphae. Conidiophores: macronematous, erect, hyaline to slightly yellow, 1–2-septate, smooth-walled to verrucose (especially at the base), 20–30 × 2–5 µm, slightly tapering toward the fertile apex, which consists in a verticillate arrangement of 5 to 15 fertile branches. Fertile branches: single, hyaline, smooth- and thin-walled, cylindrical but rounded at the end, up to 20 µm in length, 1–1.5 µm wide, forming septa basipetally to produce 3 to 5 arthroconidia released via schizolythic secession. Arthroconidia: hyaline, smooth- and thin-walled, 2–4 µm × 1.25–2 µm. Chlamydospores and sexual morphology: not observed.

Colonies on MEA: reaching 6–7 mm diameter after 2 weeks at 25 °C, velvety, slightly elevated, margins regular, orange white (5A2) at center, and white (3A1) to the edge, exudate absent, sporulation sparse; reverse pale yellow (4A3), diffusible pigment absent. Colonies on tap water agar (TWA) with sterile cork: reddish yellow (4A6), exudate absent, sporulation abundant; reverse orange (6B7), diffusible pigment absent.

Other specimens examined: Spain: Barcelona province: Sant Sadurní d’Anoia, from a sparkling wine sample, 03 June 2011, J. F. Cano-Lira & A. M. Stchigel (FMR 16677).

Notes: Despite *S. nodosa* remains as incertae sedis, a Blast search using the ITS sequence available at the GenBank/EMBL databases (data not shown) has placed this fungus phylogenetically close to members of the order Helotiales (class Leotiomycetes), a taxon phylogenetically far from the order Onygenales (class Eurotiomycetes), where the genus *Dactylodendron* is located.

***Dactylodendron pluriseptatum* Rodr.-Andr., Cano et Stchigel, sp. nov**. MycoBank MB 827863. (Figure 9)

Etymology: From Latin pluri-, many, and -septatum, septate, due to the presence of many septa along the fertile branches.

Diagnosis: Characterized by the production of successively branched conidiophores, whose verticillate arrangement of fertile branches develop long chains of disarticulating, prismatic arthroconidia. These conidiophores are similar to those of *D. pinicola*, but *D. pluriseptatum* never produce conidiomata, which is seen in *D. pinicola*.

Type: Spain: Barcelona province: Sant Sadurní d’Anoia, from sparkling wine, 03 June 2011, J. F. Cano-Lira & A. M. Stchigel (CBS H-23374 − holotype; CBS 143846 = FMR 16678 − ex-type cultures; LSU sequence GenBank LT985882).

Description: Colonies on PDA: reaching 9–10 mm diameter after 2 weeks at 25 °C, floccose, slightly elevated, margins slightly irregular, orange (5A7) at center and light orange (5A5) at edge, exudate absent, sporulation abundant; reverse deep orange (5A8), diffusible pigment absent. Mycelium: composed of septate, hyaline, smooth- and thin-walled, 1–1.5-µm wide hyphae. Conidiophores: semi-macronematous, erect, hyaline, smooth- and thin-walled, 50–75 × 1–1.5 µm, septate, repeatedly branched, fertile branches in terminal verticillate arrangements of 2–5. Fertile branches: hyaline, thin- and smooth-walled, cylindrical but rounded at the end, 15–45 × 1–1.5 µm, producing septa basipetally for delimitation of arthroconidia, which are very late released in chains or individually from the fertile branch via schizolythic secession. Arthroconidia: in long chains (up to 15 elements), hyaline, smooth- and thin-walled, cylindrical to almost cuboid, 2–4 µm × 1–1.5 µm. Chlamydospores: hyaline, one-celled, smooth- and thick-walled, irregularly globose, 5–7 µm diameter, arising laterally on the vegetative hyphae. Sexual morphology: not observed.

Colonies on MEA: reaching 9–10 mm diameter after 2 weeks at 25 °C, floccose, elevated, margins irregular, pale yellow (4A4), exudate absent, sporulation sparse; reverse greyish orange (5B6), diffusible pigment absent. Colonies on TWA with sterile cork: hairy brownish orange (6C8), exudate absent, sporulation abundant; reverse orange (6B8), diffusible pigment absent.

#### 3.4.3. Subclade C: Chaetothyriales

Because our strain FMR 16667 was placed within subclade C (Figure 1) corresponding to the species of the genus *Cladophialophora*, in a terminal branch (Figure 1 and Figure 4) together with *Cladophialophora mycetomatis*, and because FMR 16667 displays enough phenotypic and phylogenetic differences from the latter and from other species of the genus, we propose *Cladophialophora recurvata* as a new species, described as follows.

***Cladophialophora recurvata* Rodr.-Andr., Cano et Stchigel, sp. nov**. MycoBank MB 830605. (Figure 10)

Etymology: From Latin recurvatis, recurved, because the presence of coiled hyphae.

Diagnosis: Forming a terminal clade with *C. mycetomatis*, species placed within clade II of Cladophialophora s. str. (Figure 1) [51]. *Cladophialophora recurvata* differs from the latter by its production of bigger conidia (4–8 × 3–7 μm versus 2.5–4 × 2–3 μm), which are also broadly ellipsoidal to subglobose in the former and fusiform to broadly fusiform in *C. mycetomatis*, and by the inconspicuous flattened scars (much more evident in the rest of the species of the genus).

Type: Spain: Barcelona province: Sant Sadurní d’Anoia, from sparkling wine, 3 Apr. 2011, J. F. Cano-Lira & A. M. Stchigel (CBS H-23380 − holotype; CBS 143843 = FMR 16667 − ex-type cultures; LSU sequence GenBank LT985879).

Description: Colonies on MEA: reaching 21–22 mm diameter after 2 weeks at 25 °C, flat, felted, regular margins, greyish brown (5E3) at center, and olive (2F3) to the edge, exudate absent, sporulation abundant; reverse olive (2F3), diffusible pigment absent. Mycelium: composed by septate, smooth- and thin-walled, pale olivaceous hyphae, 3–4 μm wide, locally forming abundant coils from which arise most of the conidiophores. Conidiophores: micronematous, indistinguishable from the vegetative hyphae. Conidiogenous cells: mono- to polyblastic, determinate, integrated to the hyphae or discrete, in this case, ampuliform to barrel-shaped, 5–8 × 4–5 μm, arising from hyphae or the coils. Conidia: holoblastic, one-celled, pale olivaceous to pale olivaceous-brown, thin- and smooth-walled to verrucose, broadly ellipsoidal to subglobose, 4–8 × 3–7 μm, disposed in long, branched acropetal chains, with one or two inconspicuous, flattened scars of up to 3-μm wide, of the same color than the rest of the conidium; ramoconidia one-celled, cylindrical to nearly so, one-celled, in chains up to 3, 5–15 × 2–3 μm. Budding cells, chlamydospores, muriform cells, synanamorph, and sexual morphology: not observed.

Colonies on PDA: reaching 21–22 mm diameter after 2 weeks at 25 °C, flat, regular margins, greyish brown (5E3) at center, and olive (2F3) to the edge, exudate absent, sporulation abundant; reverse olive (2F3), diffusible pigment absent. Colonies on OA: reaching 20–21 mm diameter after 2 weeks at 25 °C, flat, felted, regular margins, olive grey (3D2) at center, and olive (2F3) at edge, exudate absent, sporulation sparse; reverse olive (2F3), diffusible pigment absent. Colonies on TWA with sterile cork: olive brown (4F3), exudate absent, sporulation abundant; reverse olive brown (4F3), and diffusible pigment absent.

#### 3.4.4. Subclade D: Pleosporales

Because FMR 15668 and FMR 16668, and FMR 16665 and FMR 16666, were placed within two independent sister branches (100% BS / 1 PP each one) in one terminal clade (81% BS / 0.98) of the sister clade D1 (Figure 1), and because these strains display enough morphological differences with respect to the other species of the genus, we propose the erection of the new species *Kirschsteiniothelia ebriosa* and *Kirschsteiniothelia vinigena*, which are described as follows:

***Kirschsteiniothelia ebriosa* Rodr.-Andr., Cano et Stchigel, sp. nov**. MycoBank MB 830603. (Figure 11)

Etymology: From Latin ebrios, drunk, due to the habitat of this fungus.

Diagnosis: Resembling more a species of the genus *Diplococcium* than a *Kirschsteiniothelia*’s asexual morphology (= Dendryphiopsis), because of the acropetal chains of 1-septate conidia produced laterally along the conidiophore. (These are multiseptate and solitary on the branches arising at the top of the conidiophore in *Kirschsteiniothelia*.) However, *K. ebriosa* falls within the same clade as other *Kirschsteiniothelia* spp., which is placed far away from *Diplococcium* spp. (data not shown). *Kirschsteiniothelia ebriosa* is distinguishable from all other species of the genus by its production of (mostly) 1-septate small conidia in branched acropetal chains, laterally on the stipe of the conidiophore.

Type: Spain: Barcelona province: Sant Sadurní d’Anoia, from sparkling wine, 03 June 2011, J. F. Cano-Lira & A. M. Stchigel (CBS H-23379 − holotype; CBS 143842 = FMR 16666 − ex-type cultures; LSU sequence GenBank LT985884).

Description: Colonies on MEA: reaching 28–31 mm diameter after 2 weeks at 25 °C, floccose, slightly elevated, regular margins, yellowish brown (5E4) at center, and blackish-olive (2G6) at edge, exudate absent, abundant sporulation; reverse blackish olive (2G6), diffusible pigment absent. Mycelium: abundant, composed of dark brown, septate, smooth-and thin-walled, 4-µm wide hyphae. Conidiophore: macronematous, consisting of a straight or slightly sinuous, erect, dark brown, septate, thin- and smooth-walled to slightly verrucose stipe, 40–150 × 4 µm, bearing a few lateral branches and occasionally one branch at the top; branches are brown, thin- and smooth-walled, 1–5-septate, cylindrical, 30–50 × 4 µm, with rounded ends. Conidiogenous cells: mono- to polytretic, integrated to the stype, to the branches and to the conidia, intercalary or terminal, determinate, cylindrical. Conidia: holoblastic, brown to dark brown, thin- and smooth-walled, 1–2(–5)-septate, sometimes solitary, mostly in branched acropetal chains of up to 5, cylindrical with rounded ends, 8–14 × 4–5 µm, sometimes slightly constricted at septum, produced laterally and terminally on the stipe, on the branches and on the conidia. Chlamydospores and sexual morphology: not observed.

Colonies on PDA: reaching 26–28 mm diameter after 2 weeks at 25 °C, floccose, slightly elevated, regular margins, pitch black (5H2) at center, grey (8E1) to the edge, exudate absent, sporulation sparse; reverse blackish olive (2G6), diffusible pigment absent. Colonies on OA: reaching 20–21 mm diameter after 2 weeks at 25 °C, floccose, slightly elevated, regular margins, black (18G2) at center, grey (5E1) to the edge, exudate absent, sporulation abundant; reverse blackish violet (15G8), diffusible pigment absent. Colonies on TWA with sterile cork after 2 weeks at 25 °C: blackish olive (2G6), exudate absent, sporulation abundant; reverse blackish olive (2G6), diffusible pigment absent.

Other specimens examined: Spain: Barcelona province: Sant Sadurní d’Anoia, from sparkling wine, 03 June 2011, J. F. Cano-Lira & A. M. Stchigel (FMR 16665).

***Kirschsteiniothelia vinigena* Rodr.-Andr., Cano et Stchigel, sp. nov**. MycoBank MB 830604. (Figure 12)

Etymology: From Latin vinum, wine, because the origin of the fungus.

Diagnosis: Distinguished from other species of the genus because of its production of ornamented (verrucose) stipes, branches and conidia (smooth-walled or nearly so for the other of the species), and by the production of complex system of branches (not reported for the other species).

Type: Spain: Barcelona province: Sant Sadurní d’Anoia, from cork stopper, 03 June 2011, J. F. Cano-Lira & A. M. Stchigel (CBS H-23378 − holotype; CBS 143837 = FMR 15668 − ex-type cultures; LSU sequence GenBank LT985883).

Description: Colonies on MEA: reaching 23–24 mm diameter after 2 weeks at 25 °C, floccose, slightly elevated, regular margins, blackish grey (2G1) at center, blackish olive (2G6) at the middle part, and grey (5E1) at the edge, exudate absent, sporulation sparse; reverse blackish red (10H8), diffusible pigment absent. Mycelium: abundant composed of brown, smooth- and thin-walled, septate hyphae, 2–3 µm wide. Conidiophore: macronematous, consisting in a straight or slightly sinuous, erect, dark brown, septate, thin- and smooth-walled to coarsely verrucose (specially at the base) stipe, 100–150 × 3 µm, bearing several lateral branches, which also branch irregularly; branches are abundant, brown, thin- and smooth-walled to coarsely verrucose (specially at the base), non-septate to 7-septate, barrel-shaped to cylindrical, 5–80 × 4 µm, with rounded ends. Conidiogenous cells: mono- to polytretic, integrated to the stype, to the branches and to the conidia, intercalary or terminal, determinate, and cylindrical. Conidia: holoblastic, dark brown, thin- and smooth-walled to coarsely verrucose, 1–2(–7)-septate, sometimes solitary, mostly disposed in branched acropetal chains of up to 4, cylindrical to scolecoid, 8–80 × 4–5 µm, with rounded ends, sometimes slightly constricted at septa, produced laterally and terminally on the stipe, on the branches, and on other conidia. Chlamydospores and sexual morphology: not observed.

Colonies on PDA: reaching 24–25 mm diameter after 2 weeks at 25 °C, floccose, slightly elevated, regular margins, pitch black (5H2) at center, with consecutive blackish olive (2G6) and grey (5E1) rings, and pitch black (5H2) at the edge, exudate absent, sporulation sparse; reverse blackish olive (2G6), diffusible pigment absent. Colonies on OA: reaching 14–15 mm diameter after 2 weeks at 25 °C, floccose, slightly elevated, regular margins, ink black (18G2) at center, grey (5E1) to the edge, exudate absent, sporulation sparse; reverse blackish olive (2G6), diffusible pigment absent. Colonies on TWA: with sterile cork, blackish olive (2G6), exudate absent, sporulation sparse; reverse blackish-olive (2G6), diffusible pigment absent.

Other specimens examined: Spain: Barcelona province: Sant Sadurní d’Anoia, from sparkling wine, 03 June 2011, J. F. Cano-Lira & A. M. Stchigel (FMR 16668).

## 4. Discussion

*Candida patagonica* was the only yeast retrieved in our study. This fungus has previously been reported from fermentation vats and oak barrels in the cellars of North Patagonia, Argentina [52]. Despite the ascomycetous yeasts such as *Dekkera bruxellensis*, *Hanseniaspora uvarum*, *Issatchenkia orientalis*, *Metschnikowia pulcherrima*, and some species of the genera *Candida* and *Zygosaccharomyces* [53,54] having been involved in the deterioration of wines, *C. patagonica* has not been reported as a spoilage organism for these sorts of alcoholic beverages.

The isolates belonging to the order Eurotiales showed the broadest fungal diversity, being distributed among the genera *Aspergillus*, *Penicillium*, *Rasamsonia* and *Talaromyces*. This latter genus was the most frequently recovered from both sorts of substrata, and two of them, particularly from sparkling wine, are new species, i.e., *Talaromyces speluncarum*, characterized by mostly biverticillate conidiophores and brown, spinose to verrucose, globose conidia, and *Talaromyces subericola*, which grows faster than *T. speluncarum* and produces smooth-walled conidia (spinose to verrucose in *T. speluncarum*). *Aspergillus*, another common genus in sparkling wine, was represented by *A. aureolatus*, *A. jensenii*, and *A. puulaauensis*, all of them pertaining to the section *Nidulantes* [55]. *Aspergillus aureolatus* [56] was originally isolated from air in Serbia, *A. jensenii* [57] from soil in the USA, and *A. puulaauensis* [57] from dead hardwood in the Hawaiian archipelago. There have not been any reports of these three species found in wines. *Penicillium corylophilum* [58] was isolated from sparkling wine samples. This taxon was reported mostly in damp buildings in North America and West Europe, but also from foods and from mosquitoes [59,60], vineyards, grape must, fermentation wine, and fruit juices [11,61,62]. Previously, we had isolated this fungus from the environment of the cellars where the bottles containing the sparkling wine were aging (data not published). Consequently, finding *P. corylophilum* might be due to the bottle not being sufficiently sealed by the cork stopper. A new species of *Rasamsonia*, *Rasamsonia frigidotolerans*, was found in the wine samples. The genus is characterized by the production of ornamented, paecilomyces-like conidiophores and olive-brown conidia, and in four of the species, the production of ascomata have been reported. *Rasamsonia* spp. have been reported in Asia, Europe, and North America, from substrata such as compost, conifer wood chips, cow dung, house dust, indoor air, piles of peat, rice straw, seed of *Piper nigrum*, soil, and human and animal clinical specimens [47,48,63,64]. None of the previous studies have reported this genus either in wine or on cork stoppers. *Rasamsonia* species are thermotolerant or thermophilic, with an optimum growth temperature above 30 °C and a maximum above 45 °C [65,66]. *Rasamsonia frigidotolerans* is characterized by the production of smooth-walled conidiophores (verrucose in all other species of the genus), by an absence of growth on CYA at 30 °C (all other species are thermotolerant), and by the production of globose conidia connected by disjunctors (absent in the rest of the species of the genus).

The new genus *Dactylodendron*, phylogenetically closely related to the order Onygenales, is characterized by its branched conidiophores and the production of chains of arthroconidia. The type species, *Dactylodendron pinicola*, is an asexual fungus previously classified phenotypically within the genus *Arthrographis*. It was originally isolated from insect galleries and from adult beetles of *lps latidens*, and of *Dendroctonus ponderosae* in *Pinus contorta* var. *latifolia* in Canada [67]. *Dactylodendron pinicola* produces conidiomata (absent in the other two species of the genus), whereas *D. pluriseptatum* produces long chains of prismatic arthroconidia, and *D. ebriosum* forms conidiophores that produce fertile branches at the apex.

In our study, we found a few isolates morphologically similar to the genus *Dendryphiopsis,* which in a polyphasic study demonstrated to be new species of *Kirschsteiniothelia*. This genus demonstrated to be phylogenetically related to the anamorphic genus *Dendryphiopsis* [68,69]. Species of *Kirschsteiniothelia* / *Dendryphiopsis* have been isolated principally from decaying wood and leaves [70,71,72], and even in freshwater habitats [73,74,75], but never from sparkling wine or cork stoppers. *Kirschsteiniothelia ebriosa* and *K. vinigena* differ from the other species of the genus by the absence of a sexual morphology, and the conidia arising in chains directly from the main axis of the conidiophore. These two species can be distinguished each from other by the number of septa and the length of the conidia (mostly two-celled and short in *K. ebriosa*, and multi-celled and long in *K. vinigena*).

We also isolated an interesting strain of *Cladophialophora* from sparkling wine. *Cladophialophora recurvata* produces aseptate, broadly ellipsoidal to subglobose, relatively large conidia, with inconspicuous flattened scars (more evident in the other species of the genus). The species of the genus *Cladophialophora* have been never reported from wine or cork.

Other species isolated during our study were *Alternaria alternata* and *Cladosporium cladosporioides*, both from cork stoppers. There are a few reports of *A. alternata* in grape must from the Priorat region in Spain and from Douro in Portugal [11,61]. The species of the genus *Alternaria* infects a broad variety of living plants, but can also be recovered from plant debris [76]. *Cladosporium oxysporum* (but not *C. cladosporioides*) has been isolated previously from cork stoppers [5]. *Cladosporium* species are found worldwide, and frequently occur as a secondary invader of necrotic parts of different sort of plants, but also are easily recovered from air, soil, textiles, and numerous other substrata [77]. *Beauveria bassiana*, a well-known entomopathogenic fungus [78], was isolated once from a sample of sparkling wine. This fungus is usually found in soil [19], but is also known to be endophytic in living plants including grapevine [79].

Because the new fungal species failed to proliferate at ethanol concentrations ≥ 10% v/v, we consider them to be from a different origin than the grape must, probably the cellar and/or the cork stoppers, despite some of them being recovered only from sparkling wine samples. We are hopeful that future studies will allow us to discover whether these fungal strains produce 2,4,6-TCA and other compounds responsible for cork taint.

## 5. Conclusions

The presence of yeasts and molds (and occasionally of bacteria) was detected in several (24 out of 39) samples of sparkling wine (Catalonian cava) affected by cork taint, with a musty, or mouldy, off-odor, and/or flavor alteration that makes the wine undrinkable. On the other hand, all negative controls (without appreciable organoleptic alteration) were free of fungi. All cork stoppers from negative controls and deteriorated wine developed fungal colonies. We isolated 27 different fungi from both substrata. Among them, we found a new genus (*Dactylodendron*) and eight new species (*Cladophialophora recurvata*, *Dactylodendron ebriosum*, *Dactylodendron pluriseptatum*, *Kirschteiniothelia ebriosa*, *Kirschteiniothelia vinigena*, *Rasamsonia frigotolerans*, *Talaromyces speluncarum* and *Talaromyces subericola*). All fungal taxa were able to grow on cork, but only at alcohol concentrations ≤ 10% v/v (which is lower than 11.5% strength of Catalan sparkling wines). We therefore conclude that the fungi present in sparkling wine were also present in turn on the cork stoppers and/or are part of the environment of the cellar. Although *Penicillium corylophylum* was found in wine samples, its presence does not represent per se a risk to the health of the consumer (this fungus is a mycotoxin producer), because all the fungi we found were unable to grow at the ethanol concentration of the sparkling wines. Future studies will allow us to find out whether these fungi form 2,4,6-TCA and/or other volatile organic compounds involved in the production of cork taint in wines.

## Figures and Tables

**Figure 1 microorganisms-08-00012-f001:**
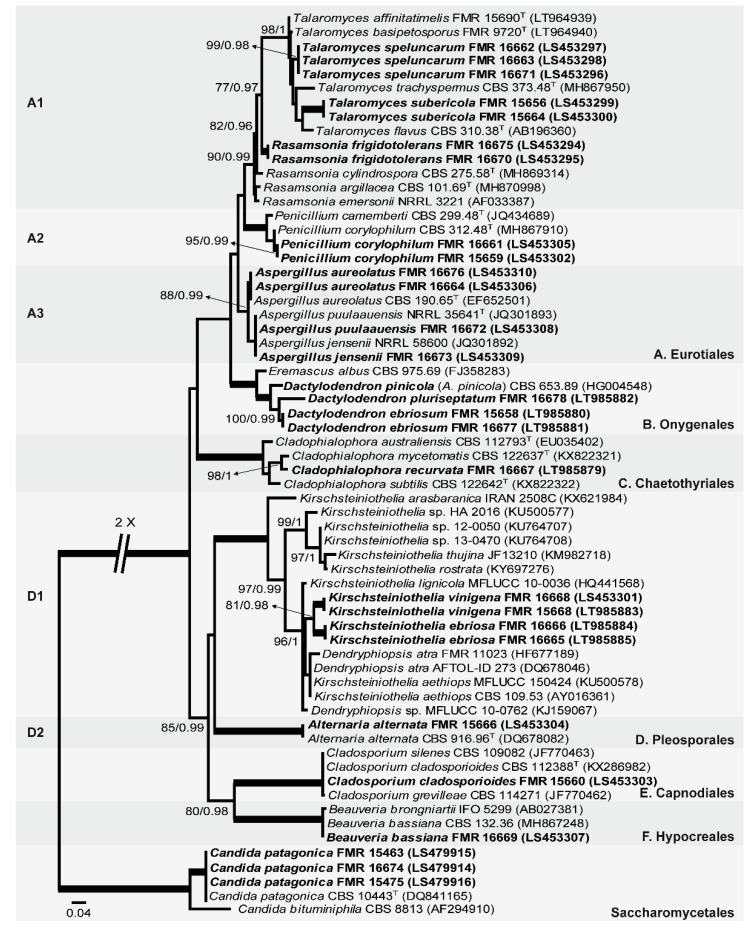
Maximum likelihood (ML) phylogenetic tree based on the analysis of LSU nucleotide sequences for all fungi isolated from sparkling wine and cork stoppers. Bootstrap support values/Bayesian posterior probability scores over 70/0.95 are indicated on the nodes. Some branches were shortened; these are indicated by two diagonal lines with the number of times a branch was shortened. ^T^ = ex type. Alignment length 519 bp. The sequences not generated by us were retrieved from EMBL/GenBank.

**Figure 2 microorganisms-08-00012-f002:**
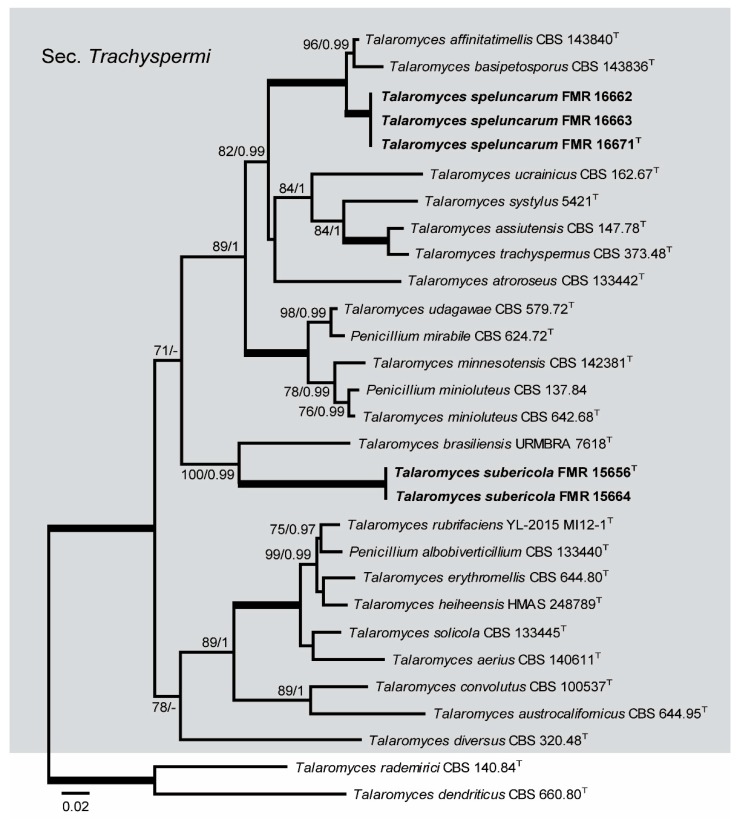
ML phylogenetic tree based on the analysis of internal transcribed spacers (ITS), *BenA*, *CaM,* and *rpb*2 concatenated dataset for species of the genus *Talaromyces* section *Trachyspermi* isolated from sparkling wine and cork stoppers. *Talaromyces rademirici* CBS 140.84 and *Talaromyces dendriticus* CBS 660.80 were chosen as out-group. ^T^ ex-type strain. Alignment length is 2270 bp. The sequences used in this analysis are in Table 1.

**Figure 3 microorganisms-08-00012-f003:**
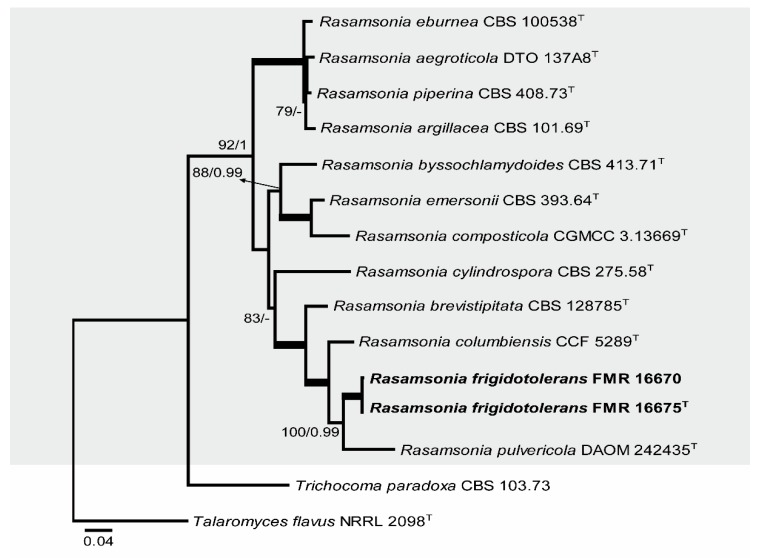
ML phylogenetic tree based on the analysis of ITS, *BenA*, and *CaM* concatenated dataset for species of the genus *Rasamsonia* isolated from sparkling wine. *Trichocoma paradoxa* CBS 103.73 and *Talaromyces flavus* NRRL 2098 were chosen as out-group. ^T^ ex-type strain. Alignment length is 1657 bp. The sequences used in this analysis are in Table 1.

**Figure 4 microorganisms-08-00012-f004:**
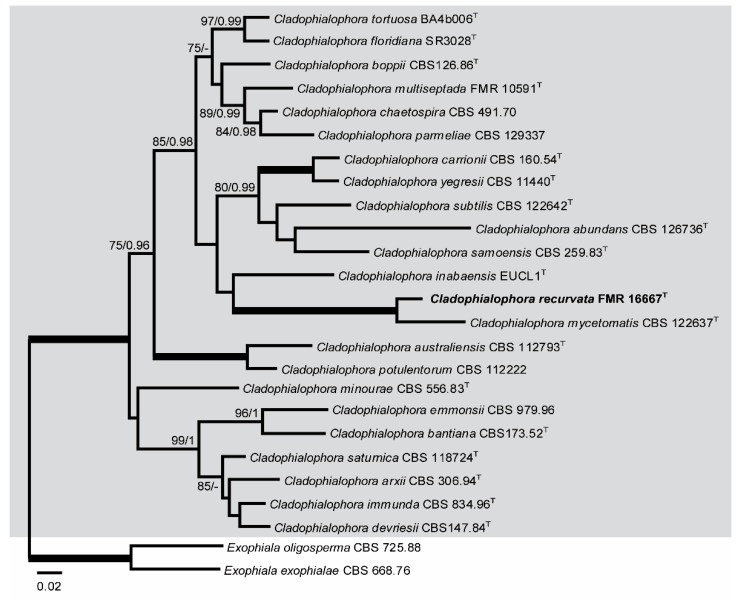
ML phylogenetic tree based on the analysis of ITS, LSU, and *BenA* concatenated dataset for species of the genus *Cladophialophora* isolated from sparkling wine. *Exophiala oligosperma* CBS 725.88 and *Exophiala exophialae* CBS 668.76 were chosen as out-group. ^T^ ex-type strain. Alignment length is 1554 bp. The sequences used in this analysis are in Table 1.

**Figure 5 microorganisms-08-00012-f005:**
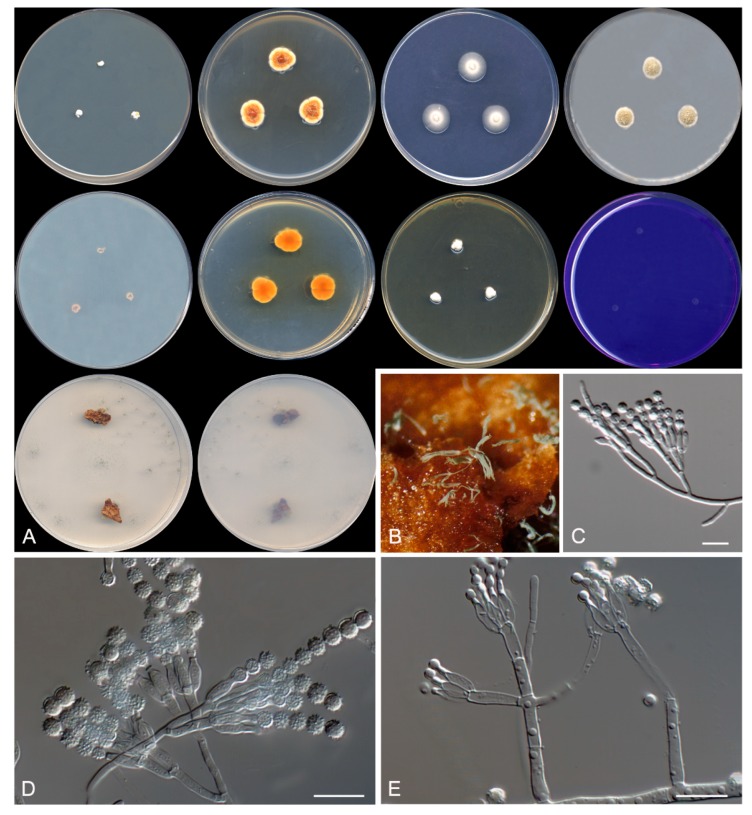
*Talaromyces speluncarum* CBS 143844. (**A**) Colonies on Czapek yeast extract agar (CYA), malt extract agar (MEA), dichloran 18% glycerol agar (DG18), and oatmeal agar (OA) after 14 days at 25°C, from left to right (top row); reverse of the colonies on CYA and MEA, and surface of the colonies on yeast extract sucrose agar (YES) and creatine sucrose agar (CREA), from left to right (medium row); surface and reverse of the colony on tap water agar (TWA) with cork, from left to right (bottom row). (**B**) Detail of the colony on TWA with cork. (**C**–**E**) Conidiophores and conidia. Scale bar = 10 μm.

**Figure 6 microorganisms-08-00012-f006:**
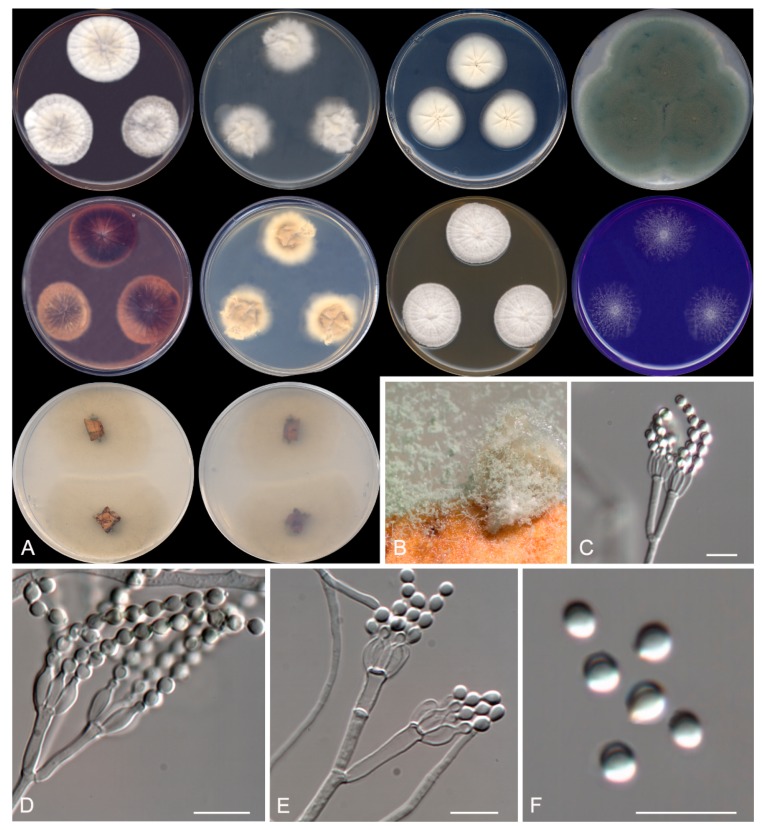
*Talaromyces subericola* CBS 144322. (**A**) Colonies on CYA, MEA, DG18, and OA after 14 days at 25°C, from left to right (top row); reverse of the colonies on CYA and MEA, and surface of the colonies on YES and CREA, from left to right (medium row); surface and reverse of the colony on TWA with cork, from left to right (bottom row). (**B**) Detail of the colony on TWA with cork. (**C**–**E**) Conidiophores. (**F**) Conidia. Scale bar = 10 μm.

**Figure 7 microorganisms-08-00012-f007:**
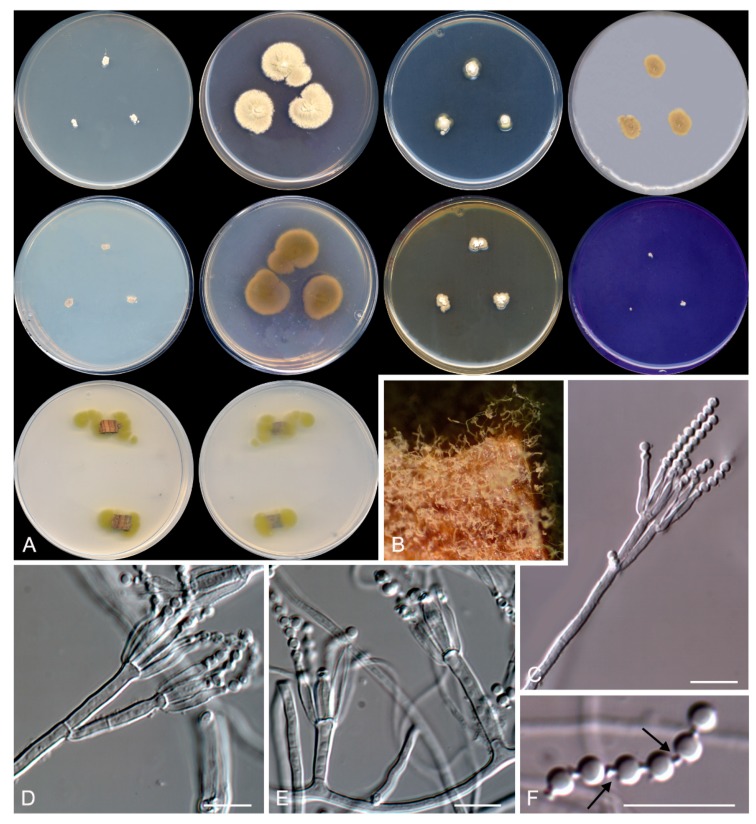
*Rasamsonia frigidotolerans* CBS 143845. (**A**) Surface of the colonies on CYA, MEA, DG18, and OA after 14 days at 25 °C, from left to right (top row); reverse of the colonies on CYA and MEA, and surface of the colonies on YES and CREA, from left to right (medium row); surface and reverse of the colony on TWA with cork, from left to right (bottom row). (**B**) Detail of the colony on TWA with cork. (**C**–**E**) Conidiophores. (**F**) A chain of conidia. The arrow shows a disjunctor between conidia. Scale bar = 10 μm.

**Figure 8 microorganisms-08-00012-f008:**
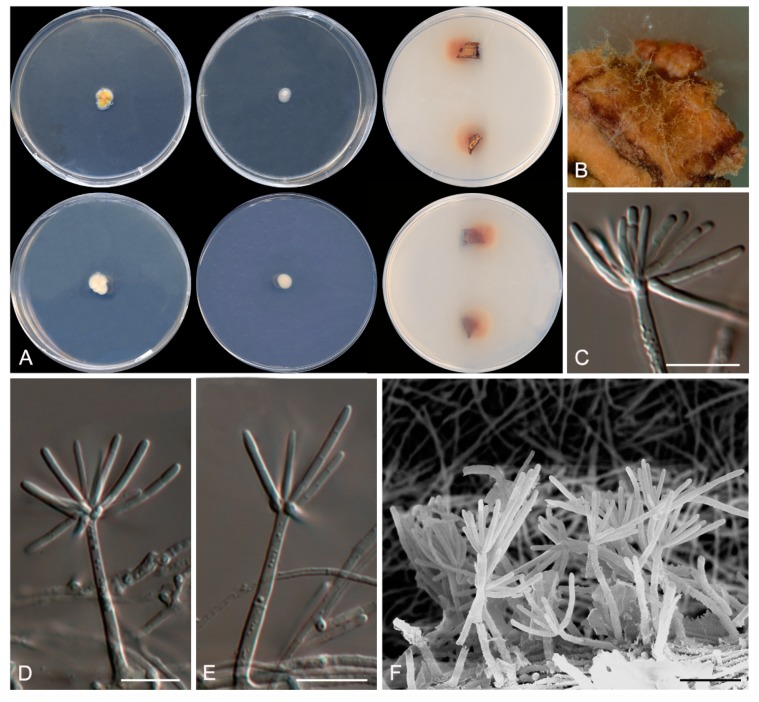
*Dactylodendron ebriosum* CBS 144321. (**A**) Colonies on potato dextrose agar (PDA), MEA, and TWA with cork after 14 days at 25 °C, from left to right (top row, surface; bottom row, reverse); (**B**) Detail of the colony on TWA with cork. (**C**–**E**) Conidiophores and conidia. (**F**) Conidiophores under SEM. Scale bar = 10 μm.

**Figure 9 microorganisms-08-00012-f009:**
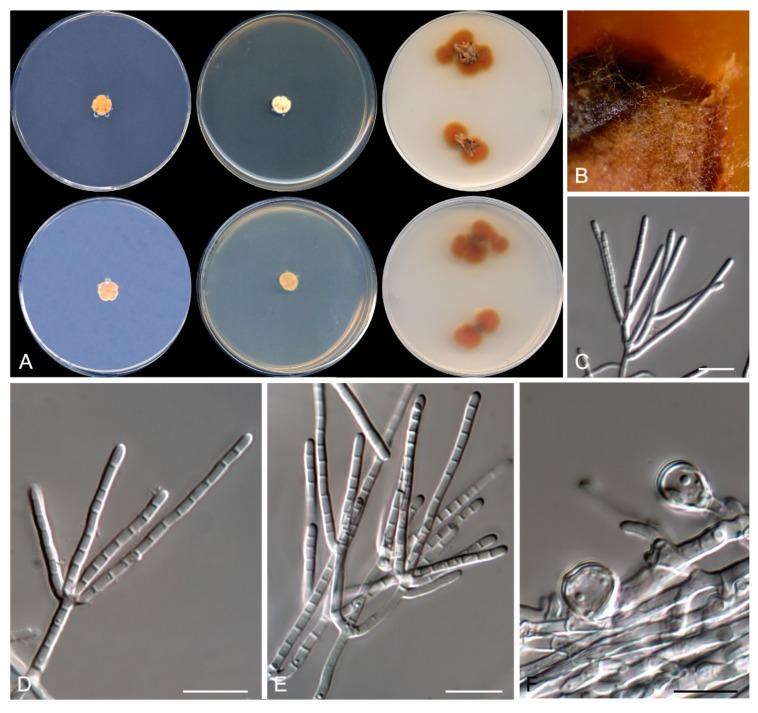
*Dactylodendron pluriseptatum* CBS 143846. (**A**) Colonies on PDA, MEA, and TWA with cork after 14 days at 25°C, from left to right (top row, surface; bottom row, reverse); (**B**) Detail of the colony on TWA with cork. (**C**–**E**) Conidiophores and conidia. (**F**) Chlamydospores. Scale bar = 10 μm.

**Figure 10 microorganisms-08-00012-f010:**
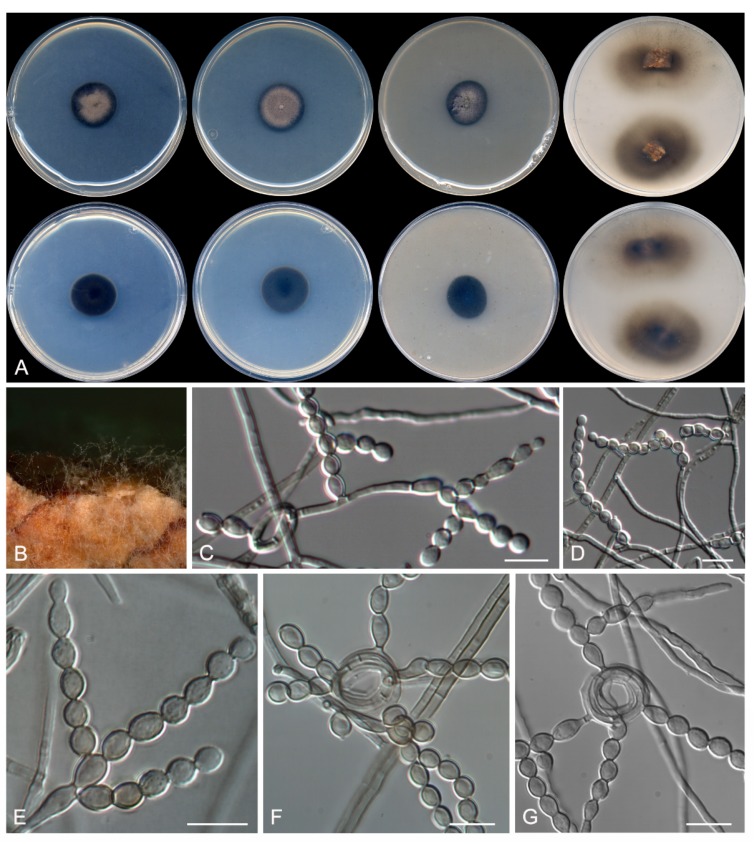
*Cladophialophora recurvata* CBS 143843. (**A**) Colonies on PDA, MEA, OA, and TWA with cork after 14 days at 25°C, from left to right (top row, surface; bottom row, reverse). (**B**) Detail of the colony on TWA with cork. (**C**–**G**) Conidiophores and conidia. Scale bar = 10 μm.

**Figure 11 microorganisms-08-00012-f011:**
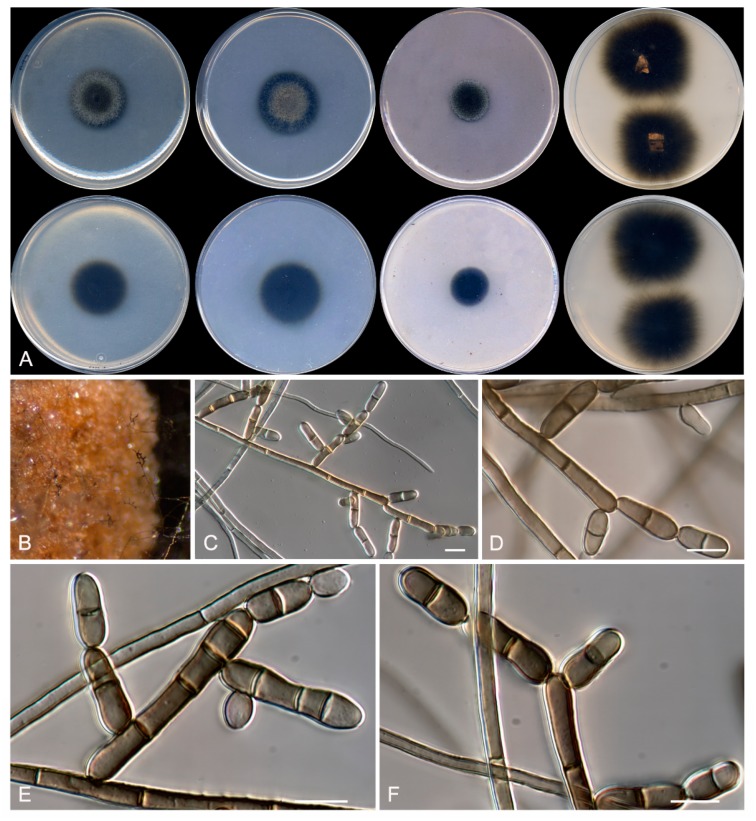
*Kirschteiniothelia ebriosa* CBS 143842. (**A**) Colonies on PDA, MEA, OA, and TWA with cork after 14 d at 25 °C, from left to right (top row, surface; bottom row, reverse). (**B**) Detail of the colony on TWA with cork. (**C**–**F**) Conidiophores and conidia. Scale bar = 10 μm.

**Figure 12 microorganisms-08-00012-f012:**
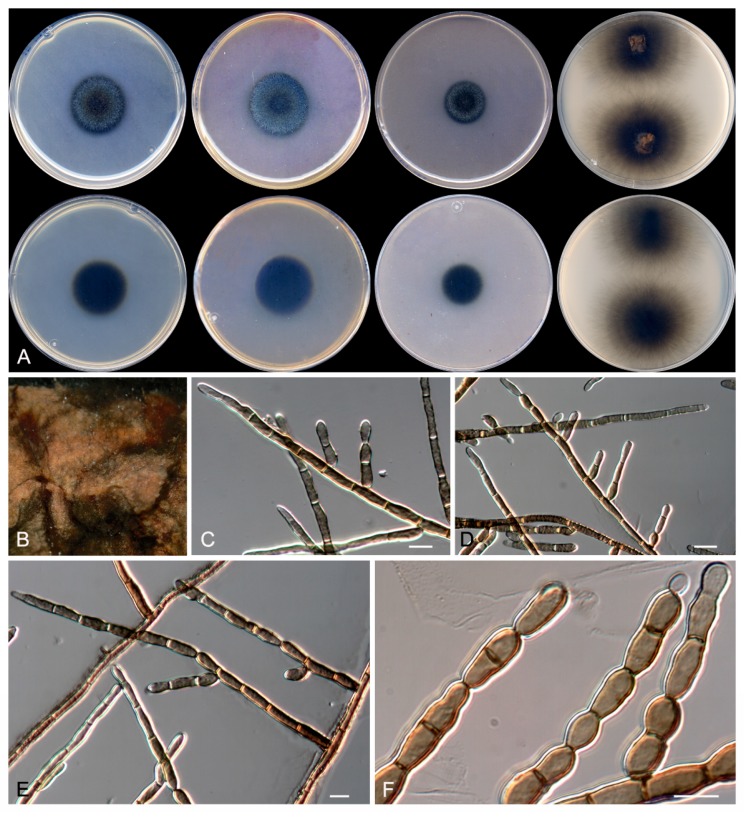
*Kirschteiniothelia vinigena* CBS 143837. (**A**) Colonies on PDA, MEA, OA, and TWA with cork after 14 days at 25 °C, from left to right (top row, surface; bottom row, reverse). (**B**) Detail of the colony on TWA with cork. (**C**–**F**) Conidiophores and conidia. Scale bar = 10 μm.

**Table 1 microorganisms-08-00012-t001:** Fungal taxa and their nucleotide sequences of the molecular markers used to build the *Cladophialophora* spp., *Rasamsonia* spp., and *Talaromyces* spp. phylogenetic trees.

Species Name	Strain	GenBank Accession #				
		*BenA*	*CaM*	*rpb*2	ITS	LSU
*Cladophialophora abundans*	CBS 126736^T^	–	–	–	KC776592	KC812100
*C. arxii*	CBS 306.94^T^	–	–	–	EU103986	KX822320
*C. australiensis*	CBS 112793^T^	–	–	–	EU035402	EU035402
*C. bantiana*	CBS 173.52^T^	–	–	–	EU103989	–
*C. boppii*	CBS 126.86^T^	–	–	–	EU103997	FJ358233
*C. carrionii*	CBS 160.54^T^	EU137201	–	–	EU137266	FJ358234
*C. chaetospira*	CBS 491.70	–	–	–	EU035405	EU035405
*C. devriesii*	CBS 147.84^T^	–	–	–	EU103985	KC809989
*C. emmonsii*	CBS 979.96	–	–	–	EU103996	–
*C. floridiana*	NRRL 66282^T^	–	–	–	AB986343	AB986343
*C. immunda*	CBS 834.96^T^	EU137203	–	–	EU137318	KC809990
*C. inabaensis*	EUCL1^T^	–	–	–	LC128795	LC128795
*C. minourae*	CBS 556.83^T^	–	–	–	AY251087	FJ358235
*C. multiseptada*	CBS 136675^T^	–	–	–	HG003668	HG003671
*C. mycetomatis*	CBS 12263^T^	–	–	–	FJ385276	KX822321
*C. parmeliae*	CBS 129337	–	–	–	JQ342180	JQ342182
*C. potulentorum*	CBS 112222	–	–	–	EU035409	EU035409
*C. samoensis*	CBS 259.83^T^	EU137174	–	–	EU137291	KC809992
*C. saturnica*	CBS 118724^T^	–	–	–	EU103984	–
*C. subtilis*	CBS 122642^T^	–	–	–	FJ385273	KX822322
*C. tortuosa*	NRRL 66284^T^	–	–	–	AB986424	AB986424
***C. recurvata***	**FMR 16667^T^**	**LT985894**	–	–	**LT985878**	**LT985879**
*C. yegresii*	CBS 114405^T^	EU137209	–	–	EU137323	KC809994
*Exophiala exophialae*	CBS 668.76	EF551499	–	–	NR111130	KX712348
*E. oligosperma*	CBS 725.88	KF928550	–	–	NR111134	KF928486
*Rasamsonia aegroticola*	DTO 137A8^T^	JX273020	JX272956	–	JX272988	–
*R. argillacea*	CBS 101.69^T^	JF417456	JF417501	–	JF417491	–
*R. brevistipitata*	CBS 128785^T^	JF417454	JF417499	–	JF417488	–
*R. byssochlamydoides*	CBS 413.71^T^	JF417460	JF417512	–	JF417476	–
*R. columbiensis*	CBS 141097^T^	LT548285	–	–	LT548281	–
*R. composticola*	CGMCC 3.13669^T^	JF970183	JQ729688	–	JF970184	–
*R. cylindrospora*	CBS 275.58^T^	JF417448	JF417493	–	JF417470	–
***R.**frigidotolerans***	**FMR 16675^T^**	**LT985895**	**LT985897**	–	**LT985886**	**LS453294**
***R.**frigidotolerans***	**FMR 16670**	**LT985896**	**LT985898**	–	**LT985887**	**LS453295**
*R. eburnea*	CBS 100538^T^	JF417462	JF417494	–	JF417483	–
*R. emersonii*	CBS 393.64^T^	JF417463	JF417510	–	JF417478	–
*R. piperina*	CBS 408.73^T^	JX273000	JX272936	–	JX272968	–
*R. pulvericola*	DAOM242435^T^	KF242520	KF242522	–	KF242514	–
*Talaromyces aerius*	CBS 140611^T^	KU866835	KU866731	KU866991	KU866647	–
*T. albobiverticillius*	CBS 133440^T^	KF114778	KJ885258	KM023310	HQ605705	–
*T. assiutesis*	CBS 147.78^T^	KJ865720	KJ885260	KM023305	N899323	–
*T. atroroseus*	CBS 133442^T^	KF114789	KJ775418	KM023288	KF114747	–
*T. austrocalifornicus*	CBS 644.95^T^	KJ865732	KJ885261	–	JN899357	–
*T. basipetosporus*	FMR 9720^T^	LT906563	–	LT906545	LT906542	–
*T. brasiliensis*	CBS 142493^T^	LT855560	LT855563	LT855566	MF278323	–
*T. convolutus*	CBS 100537^T^	KF114773	–	JN121414	JN899330	–
*T. dendriticus*	CBS 660.80^T^	JX091391	KF741965	JN121547	JN899339	–
*T. diversus*	CBS 320.48^T^	KJ865723	KJ885268	KM023285	KJ865740	–
*T. affinitatimellis*	FMR 15690^T^	LT906552	LT906549	LT906546	LT906543	–
*T. erythromellis*	CBS 644.80^T^	HQ156945	KJ885270	KM023290	JN899383	–
*T. flavus*	NRRL 2098^T^	EU021663	EU021694	–	EU021596	–
*T. heiheensi* *s*	CGMCC 3.18012^T^	KX447525	KX447532	KX447529	KX447526	–
*T. minioluteus*	CBS 642.68^T^	KF114799	KJ885273	JF417443	JN899346	–
*T. minnesotensis*	FMR 14265^T^	LT559083	LT795604	LT795605	LT558966	–
*T. mirabile*	CBS 624.72^T^	KF114797	–	–	NR138300	–
*T. rademirici*	CBS 140.84^T^	KJ865734	–	KM023302	JN899386	–
*T. rubrifaciens*	CGMCC 3.17658^T^	KR855648	KR855653	KR855663	KR855658	–
*T. samsonii*	CBS 137.84^T^	KF114798	–	–	NR138301	–
*T. solicola*	DAOM 241015^T^	GU385731	KJ885279	KM023295	FJ160264	–
***T. speluncarum***	**FMR 16671^T^**	**LT985901**	**LT985906**	**LT985911**	**LT985890**	**LS453296**
***T. speluncarum***	**FMR 16662**	**LT985902**	**LT985907**	**LT985912**	**LT985891**	**LS453297**
***T. speluncarum***	**FMR 16663**	**LT985903**	**LT985908**	**LT985913**	**LT985892**	**LS453298**
*T. systylus*	BAFCcult 3419^T^	KR233838	KR233837	–	KP026917	–
*T. trachyspermus*	CBS 373.48^T^	KF114803	KJ885281	JF417432	JN899354	–
*T. ucrainicus*	CBS 162.67^T^	KF114771	KJ885282	KM023289	JN899394	–
*T. udagawae*	CBS 579.72^T^	KF114796	–	–	JN899350	–
***T.**subericola***	**FMR 15656^T^**	**LT985899**	**LT985904**	**LT985909**	**LT985888**	**LS453299**
***T.**subericola***	**FMR 15664**	**LT985900**	**LT985905**	**LT985910**	**LT985889**	**LS453300**
*Trichocoma paradoxa*	CBS 103.73	JF417469	JF417506	–	JF417486	–

BAFCcult: Culture collection of the Department of Biological Science, Faculty of Exact and Natural Sciences, Buenos Aires, Argentina; CBS: Culture collection of the Westerdijk Biodiversity Institute, Utrecht, the Netherlands; CGMCC: China General Microbiological Culture Collection Centre, Beijing, China; DAOM: Canadian Collection of Fungal Cultures, Ottawa, Canada; DTO: Applied and Industrial Mycology Department Collection, Utrecht, the Netherlands; FMR: Faculty of Medicine Reus culture collection, Spain; NRRL: ARS Culture Collection, Peoria, United States. ^T^: ex-type strain. Sequences newly generated in this study are indicated in **bold**. ITS: internal transcribed spacer region 1 and 2 including 5.8S nrDNA; LSU: 28S large subunit of the nrRNA gene; *BenA*: β-tubulin; *CaM*: calmodulin; *rpb*2: partial RNA polymerase II, second largest subunit.

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
