# Peer review of "Fungal Diversity of Deteriorated Sparkling Wine and Cork Stoppers in Catalonia, Spain"

_microorganisms, 2019, doi:10.3390/microorganisms8010012_

Round 1
Reviewer 1 Report
The work is very interesting regarding the occurrence of fungi in sparkling wines.
First of all the language should improve since in several parts of the manuscript some words are missing.
Some specific comments:
Lines 73-81. Provide reference of the method.
Line 152 and 696. Bacterial or yeast colonies. Explain how they were detected. There is nothing in materials regarding bacterial and yeast detection.
Line 158. All of them
Line 648. Was found in the wine samples
Line 701. “some of them never reported previously as altering the organoleptic properties of (sparkling) wines”. Be careful. You are not sure that these fungi are responsible for the off-odor. No such experiments were contacted. Please revise.
Author Response
Dear Reviewer,
First of all the language should improve since in several parts of the manuscript some words are missing.
RESPONSE: The manuscript has been corrected by a native English teacher
Some specific comments:
Lines 73-81. Provide reference of the method.
RESPONSE: The techniques we used for fungal isolation were not taken from previous studies as most of them were quantification techniques. In our case, we used two general microbiological techniques for fungal isolation from both sorts of substrates.
Line 152 and 696. Bacterial or yeast colonies. Explain how they were detected. There is nothing in materials regarding bacterial and yeast detection.
RESPONSE: An explanation of bacterial and/or yeast colony detection has been added in Materials and Methods, in the “Fungal isolation” paragraph (2.1.).
Line 158. All of them
RESPONSE: Changes made.
Line 648. Was found in the wine samples
RESPONSE: Changes made.
Line 701. “some of them never reported previously as altering the organoleptic properties of (sparkling) wines”. Be careful. You are not sure that these fungi are responsible for the off-odor. No such experiments were contacted. Please revise.
RESPONSE: The conflictive part of this sentence has been removed.
Sincerely,
The authors

Reviewer 2 Report
Dear authors
Please revise your manuscript according to the comments:
- please check italic for species names throughout all the text.
- how the threshold of 98% was chosen for LSU species discrimination?
- which threshold was used for the other marker used?
The paper is very detailed description of several species of filamentous fungi, some of them reported for the first time. Maybe it is a bit long and the authors should fid a way to should find a way to shorten it a little.
Author Response
Dear Reviewer,
- please check italic for species names throughout all the text.
RESPONSE: All scientific names, as well as those of the phylogenetic markers, were italicized.
- how the threshold of 98% was chosen for LSU species discrimination?
RESPONSE: It is of general acceptance among mycologists that the similarity between the nucleotide sequences of phylogenetically informative molecular markers (including 28S rDNA) is less than 98% for two different species of the same genus.
- which threshold was used for the other marker used?
RESPONSE: A value less than 98% of similarity between nucleotide sequences was also used to discriminate two different species, regardless of the sort of the molecular marker. Consequently, the sentence was reordered.
- The paper is very detailed description of several species of filamentous fungi, some of them reported for the first time. Maybe it is a bit long and the authors should fid a way to should find a way to shorten it a little.
RESPONSE: We thank the suggestion of shortening the length in order to improve the quality of the article. However, the authors consider that the chapters have the appropriate extension to reproduce the results presented herein. Regarding the description of the new taxa, its extension is within the general accepted standards.
Sincerely,
The authors